# Mathematical Modelling of Temperature Distribution in Selected Parts of FFF Printer during 3D Printing Process

**DOI:** 10.3390/polym13234213

**Published:** 2021-12-01

**Authors:** Tomáš Tichý, Ondřej Šefl, Petr Veselý, Karel Dušek, David Bušek

**Affiliations:** 1Department of Electrotechnology, Czech Technical University in Prague, Technická 2, 166 27 Praha, Czech Republic; veselp13@fel.cvut.cz (P.V.); dusekk1@fel.cvut.cz (K.D.); busekd1@fel.cvut.cz (D.B.); 2Department of Electrical Power Engineering, Czech Technical University in Prague, Technická 2, 166 27 Praha, Czech Republic; seflondr@fel.cvut.cz

**Keywords:** FFF, fused filament fabrication, additive manufacturing, temperature distribution modelling, dynamic and static model, thermal history

## Abstract

This work presented an FEM (finite element method) mathematical model that describes the temperature distribution in different parts of a 3D printer based on additive manufacturing process using filament extrusion during its operation. Variation in properties also originate from inconsistent choices of process parameters employed by individual manufacturers. Therefore, a mathematical model that calculates temperature changes in the filament (and the resulting print) during an FFF (fused filament fabrication) process was deemed useful, as it can estimate otherwise immeasurable properties (such as the internal temperature of the filament during the printing). Two variants of the model (both static and dynamic) were presented in this work. They can provide the user with the material’s thermal history during the print. Such knowledge may be used in further analyses of the resulting prints. Thanks to the dynamic model, the cooling of the material on the printing bed can be traced for various printing speeds. Both variants simulate the printing of a PLA (Polylactic acid) filament with the nozzle temperature of 220 °C, bed temperature of 60 °C, and printing speed of 5, 10, and 15 m/s, respectively.

## 1. Introduction

3D printing (also known as Additive Manufacturing (AM)) has become widespread among industry; small series production and prototyping greatly benefits from its simplicity and especially versatility. AM can also be used to produce location-specific material compositions (heterogenous parts), also in terms of their structure [1], but the technique that has become widespread among general public recently are 3D printers based on polymer material extrusion. They are often referred to as Fused Filament Fabrication (FFF) technology and are popular since they usually have simple controls and are affordable. Increasingly, more and more companies around the world are becoming specialized in 3D printers or 3D printing materials production. The principle of the FFF printer is based on the extrusion of melted thermoplastic material and its deposition on the printing platform (bed) layer by layer. The material is in the form of a string (filament) with a standard diameter of 1.75 mm or 3 mm. Further description is based on the design of the 3D printer MK3S from Prusa Research company (Czech Republic), which represents one of the most common types of FFF printers. They are based on the RepRap concept, which is characterized by that part of the printer components is printed by another printer. It reduces the overall cost of the printer [2].

The string is pulled by drive gear through the printing head. It consists of a heatsink that dissipates heat and ensures that the filament is not melted prematurely, and a hot-end part with an extrusion nozzle. The hot-end is separated from the heatsink by a heat-break, which is a thin metal tube. It also helps to keep the filament unmelted before the hot-end and thereby ensures smooth movement of the filament through the extrusion head [2].

Common materials used in filament extrusion printing are PET-G (polyethylene terephthalate glycol-modified), ASA (acrylonitrile styrene acrylate), or PLA (polylactic acid). In this paper, we focused on PLA, which is probably the most widespread material for 3D printing due to its good printability and biodegradability. Mechanical properties of structures printed from PLA are comparable to other mentioned materials, and it also has potential for use in electrical engineering [3].

To achieve good printing quality, proper setting of the printing process is necessary. One of the crucial process parameters is the temperature of the nozzle and the bed [4]. Our work aimed to describe the temperature distribution around these components, and especially around the printed material (printed with the layer thickness of 150 µm). Since measuring the temperature distribution during printing is nearly impossible, we created a two-dimensional FEM (finite element method) mathematical model.

## 2. Models and Inputs

There have been numerous works dealing with thermal simulation of the 3D printing, sometimes with a validation on a real print [5], sometimes with a focus on printing of specific structures and geometries [1], and often on studying the effect of interlayer interactions [6,7].

This work focused on the area that has not been sufficiently covered in the above-mentioned works, and that is the area between the extruder nozzle and the preheated bed. In contrast to [5], where the axial symmetry was anticipated, the temperature model was calculated with respect to real nozzle shape. Rotational symmetry of the nozzle would not be applicable for the nozzle-bed interface, as the bed is not a rotationally symmetrical object.

Two models, static and dynamic, were created in this work. The static model provided the temperature distribution in the given geometry; hence, the temperature in places that cannot be normally accessed (and measured) may be estimated. Such an example is the printed material in the area just below the printer nozzle. Furthermore, the distribution of spatial temperature changes in the printer material (filament) can be visualized as well. In the case of the dynamic model, these changes are also observed in time and movement. The heat conduction in both models is described by the Fourier-Kirchhoff Equation:(1)$$\rho \cdot c_{p}\cdot \left(\frac{\partial T}{\partial t}+v\cdot \nabla T\right)+\nabla \cdot q=Q_{v}$$
where the heat flux ***q*** is defined as:(2)q=λ·∇T

This form is not suitable for numerical solutions; hence, it is typically transformed into individual equations for each axis of the used geometry (2D or 3D). The terms in parenthesis are often expanded, so the temperature variable can be separated from the rest. Nevertheless, the form of Equation (1) illustrates the issue most satisfactorily. The terms *ρ* and *c*_p_ are material parameters: density and specific heat capacity, respectively. Another material parameter is the thermal conductivity *λ*. In static and stationary problems, the time and spatial changes are zero, and the temperature distribution is described only by the divergence of heat flux ***q*** and by heat source *Q*_v_. In dynamic situations, the full Equation (1) must be used to describe both time and spatial changes of temperature.

### 2.1. Static Model and Inputs

Before the realization of the dynamic model, it was deemed beneficial to first create a simplified description of the problem using a static model of temperature distribution in the area of printer head, bed, and part of the printed material. As was stated earlier, the time change of parameters in the static model is zero; hence the employed Equation (1) can be significantly reduced. Moreover, no mutual movement of any parts was present, and therefore the model was stationary (spatial change of parameters is zero) as well. For calculation of field distribution, MUMPS (MUltifrontal Massively Parallel sparse direct Solver) [8] was used. The model’s areas of interest represent the choice of substantial thermal-sensitive parts of the printer: heat block, extruder, heat sink, nozzle, printed material, and heated printer bed.

Figure 1 illustrates the modelled geometry, in which the blue lines highlight the boundary conditions, where the temperature was fixed at 220 °C.

On the edge of the heat sink (the upper part of the geometry where distinct fins are present), a boundary condition in the form of dissipated heating power into the ambient air was used. The total power dissipated from the surface was chosen as 40 W, as the resulting temperature distribution in the extruder axis corresponded to the one presented in a similar work [9] in which the airflow was modelled. No additional airflow was considered, as in this case, the printer was placed in a heated box in which the ambient temperature was practically constant. The grey area of Figure 1 represents the air domain, whose boundary condition was set to 25 °C (ambient temperature). This domain primarily serves to estimate the temperature distribution on the surface of the printed material and the nozzle. Lastly, the temperature of the printer bed was set at a constant 60 °C, which corresponds to the standard setting of PLA filament printing. All the employed boundary conditions can be seen in Table 1. Material parameters used in the simulation are listed in Table 2.

The model disregarded phase and temperature changes of these parameters for all materials. The reason is that the temperature of most parts does not practically change, except for the PLA filament and air. However, both thermal conductivity and specific heat capacity of air are very low, and hence their changes play virtually no role in the determination of temperature distribution. On the other hand, even though PLA’s parameters are non-negligible, their changes in the range of the observed temperatures (60 to 220 °C) are marginal (only about 8%) [10], and hence only a small error in the calculation occurs by disregarding these changes.

### 2.2. Dynamic Model and Inputs

During the printing process, the printer head performs various movements at different speeds. As a result, the temperature distribution in the printed material is different from the static case and changes according to the movement of the printer head. Hence, a dynamic model, which simulates the movement of the printer head and partially even the extrusion process, was created. Due to the dependence of parameters on both time and position, Equation (1) must be considered in its full form. For time-dependent calculation, the implicit BDF (Backward Differentiation Formula) solver with variable time stepping was used, while the MUMPS solver was employed for auxiliary static field distribution calculations.

Due to the complex nature of the simulation, only the geometry of the printer nozzle and bed was modelled (i.e., parts crucial in the extrusion and printing process). Modeling of the heat sink would increase the calculation time considerably, and its rather intricate surface could cause issues in mesh generation during the movement of the printer head. Moreover, the heat sink part does not influence the temperature distribution in the area of interest (nozzle and bed) and can thus be disregarded. The overall geometry used in the dynamic model is shown in Figure 2, which also includes the calculation mesh.

The mesh changes with each step of the calculation, and therefore the mesh in the area where the nozzle movement occurs was chosen as square shaped. The total number of elements remains constant; therefore, when the movement occurred and the nozzle in Figure 2 moved to the right, then the mesh before the nozzle (grid on the right side) condensed and the mesh located after the nozzle (in the figure on the left side) became loose.

The maximum mesh element size for the final calculation was set to 0.05 mm in the printed material (blue area on Figure 2). The calculation precision was of utmost priority in this area, as the required temperature distribution was obtained here. The impact of the mesh elements’ volume in this area was evaluated carefully before the final solving; convergence of the results depending on the mesh quality was observed.

For this purpose, the temperature in the printed material under the nozzle was evaluated. The evaluation was carried out on the cut line in the middle of the height of printed material, and results for different mesh qualities can be seen in Figure 3.

For results solved with coarse elements, we can see the different course (and peak value) of the temperature than for the results obtained with detailed meshing. When elements with a maximal dimension of 1 mm were used, the results differed by about 6%, whereas with 0.1 mm, the differences were not larger than 0.5% (both compared to results with max. element size 0.05 mm). Even though the latter maximal element size model returned satisfactorily accurate results, the triangle element maximal size of 0.05 mm was set for the final calculation for good measure. A summary of the mesh element size parameters is presented in Table 3 and includes the corresponding model computational time.

The geometry includes an array of auxiliary construction parts, which are vital for the correct function of various parts of the model (movement simulation, suitable meshing). An example of these parts is the rectangular block on the right-hand side, which consists of quadrilateral elements. Such elements are much more suitable for horizontal movement than triangular elements, which would require continuous reconstruction during the movement.

The model simulated the movement of the printer head above the printer bed. Three different speeds were examined—5 mm/s, 10 mm/s, and 15 mm/s (see Table 4)—resulting in three different temperature distributions. The other parameters and conditions were identical to the static model.

Since the model is time-dependent, initial conditions must be set as well. Naturally, they were based on the results of the static model. The temperature of the printer nozzle and bed was constant in time, as it is regulated, whereas the temperature distribution of the printed material changed due to the movement of the printer head.

In order to track the influence of the printer bed’s heat capacity on the cooling process of the printed material, the constant temperature area (i.e., the heat source) was placed inside in bed rather than on its surface. In this manner, the local cooling processes of the printer bed can take place. Such setup corresponds to the actual configuration of a 3D printer, where a sheet is placed on the heated printer bed. Due to the static nature of air in this model, distinct areas with heightened temperature will be present in the movement path of the printer head.

Ultimately, the dynamic model should provide us with the distribution of temperature in time in the area inside and outside the extruder. Therefore, the thermal history of the printed material during the complete printing process can be estimated.

## 3. Results

### 3.1. Results from the Static Model

The temperature distribution obtained from the static model is shown in Figure 4. The temperature of the upper part of the heat sink reached about 60 °C, which corresponds to the temperature given in [9]. This value is determined by the dissipated power leaving the extruder surface. As stated earlier, this power was equal to 40 W.

If we focus on the area near the heat block, it is evident that the temperature just below the heat break started to grow. The temperature started at about 100 °C in the upper part of the break and then reached a value of 220 °C as it entered the inner part of the heat block. On the other side of the heat block, the printed material exited via the nozzle and quickly cooled down to a temperature of 60 °C, i.e., the temperature of the printer bed.

This process is better illustrated in Figure 5, which shows the detail of the nozzle/bed interface (left) together with the temperature trend of the material along the deposited layer axis (right). However, the initial part of the axis is situated in the air (prior to the extrusion and deposition)—it is therefore shown as a dashed part of the resulting curve.

### 3.2. Results from the Dynamic Model

As in the previous case, the dynamic model provides knowledge of temperature distribution in the given geometry. However, the distribution is now time-dependent, as the printer head performs the movement. Figure 6 shows the temperature distribution in and around the heat block at time *t* = 5 s with a simulated movement speed of 5 mm/s.

A thin white line representing the axis of the filament/deposited layer can be observed. Along this line, the thermal history of the material is examined, as illustrated in Figure 7.

Therein, the dashed line represents the movement of the printed material through the extruder. As the material exited the extruder via the nozzle, the line became solid.

A distinct “tooth” in the temperature trend was present—this phenomenon is caused by the subsequent movement of the printer head, during which the hot brass edge of the nozzle passes the material closely, increasing its temperature once again.

Furthermore, the temperature of the material on the printer bed can be examined for various times t. The temperature trend was obtained by placing a singular point above the printer bed in the middle of the imaginary layer that would be printed and 7 mm ahead of the nozzle. The temperature of this point was calculated for each step of the printer head movement. Figure 8 shows the results of these calculations for speeds 5 mm/s (blue line), 10 mm/s (green line), and 15 mm/s (red line).

The slopes of the decreasing part of the temperature trends in Figure 8 were calculated in order to compare the individual cooling processes. The slope was considered from the second peak (i.e., from the “tooth”) onwards, and was determined as the slope of a line passing through the curve intersection with 10% and 90% of the difference between the maximum and minimum temperatures. This is clearly visible from Table 5, where the temperature decline is numerically compared. This allows the evaluation of the movement speed effect. The temperature decline was initially linear with the increasing speed, but with faster speeds, the material could not absorb the heat energy and the deviation from the linear slope slightly increased. Therefore, it may be assumed that even higher speeds would further increase the temperature slope, which would significantly change the physical parameters of the print.

## 4. Discussion

As a result of the static model, the temperature distribution in the extrusion head surroundings was presented. Remarkably, the influence of the nozzle on the material on the bed was apparent. The constantly heated printing nozzle with a different heat conductivity than the printed material and its geometry caused a typical “tooth” on the spatial temperature distribution.

In the dynamic model, a significant difference in the cooling process between different speeds of the extrusion head movement was observed.

The higher the speed of the head, the higher the slope of the temperature curve. When the speed is low, the material (in our case, semicrystalline PLA) absorbs more heat energy, which may affect the physical properties of the final print. The cooling rate directly affects the degree of crystallization of the polymer melt, which has a direct impact on the content of the crystalline phase [11]. Faster cooling results in lower crystalline content. Crystalline content directly affects the physical properties of the print, especially the mechanical properties, such as the Young’s modulus of elasticity [12].

Furthermore, it is apparent from the dynamic model that the bed temperature remained constant even though the hot-end was present above the observed point. We attribute this fact to a large amount of mass of the printing bed with high specific heat capacity.

## 5. Conclusions

The presented FEM models describe the temperature distribution in the extrusion head of the polymer extrusion FFF printer and, more importantly, inside the printed material.

The results from the static model showed the temperature distribution along the path of the filament through the extrusion head. Additionally, the simulation revealed the influence of the extrusion nozzle on the layer of already printed material. The temperature inside the printed material under the nozzle could reach 110–120 °C. Such temperature exceeds the glass transition temperature of PLA and is approaching the melting point, which ranged from 130 to 220 °C [13]. It is consistent with the principle of additive manufacturing; the bottom layer is partially melted, and the top layer sticks to it.

The results from the dynamic model are represented by the time-temperature curves of the observed point lying on the printing bed over which the extrusion head passes. The simulation showed that with the increasing speed of the extrusion head, the cooling rate also significantly increased and influenced the thermal history of the printed material, which has an impact on the mechanical or thermal properties.

These simulations and results demonstrate the importance of the proper process settings and their influence on the thermal history of the material. Together with further practical experiments that would prove the relation between the thermal history of PLA and its physical properties, it would lead to a better understanding of how to set up the FFF process properly.

From the point of further improvement of the model, we will also focus on the influence of the bed temperature setting on the temperature distribution.

## Figures and Tables

**Figure 1 polymers-13-04213-f001:**
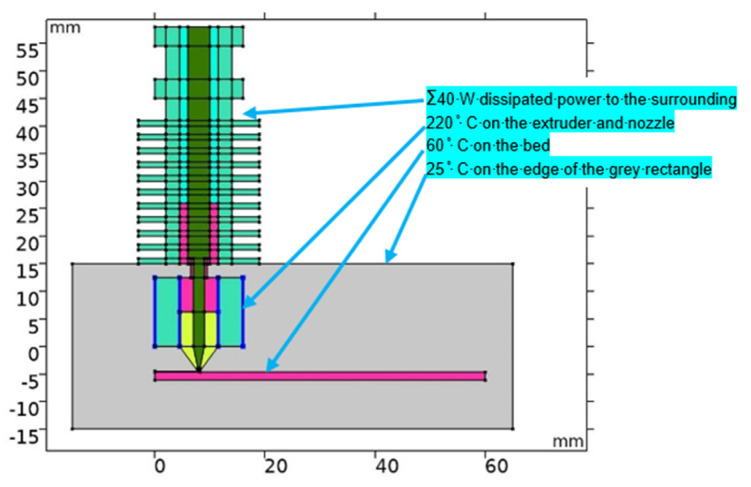
Geometry of the static model with boundary conditions.

**Figure 2 polymers-13-04213-f002:**
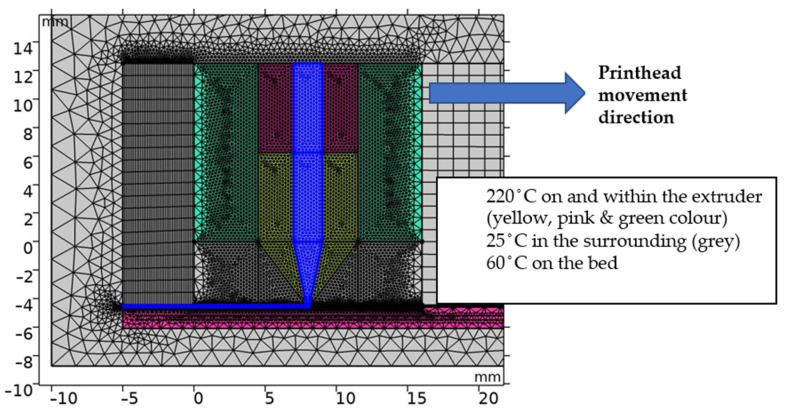
Geometry of the dynamic model with initial conditions; light turquoise—aluminum (heat block); blue—PLA (PLA material in the nozzle and on the bed); pink—steel (bed and heat break); yellow—brass (nozzle); gray—surroundings (air). The nozzle moves to the right.

**Figure 3 polymers-13-04213-f003:**
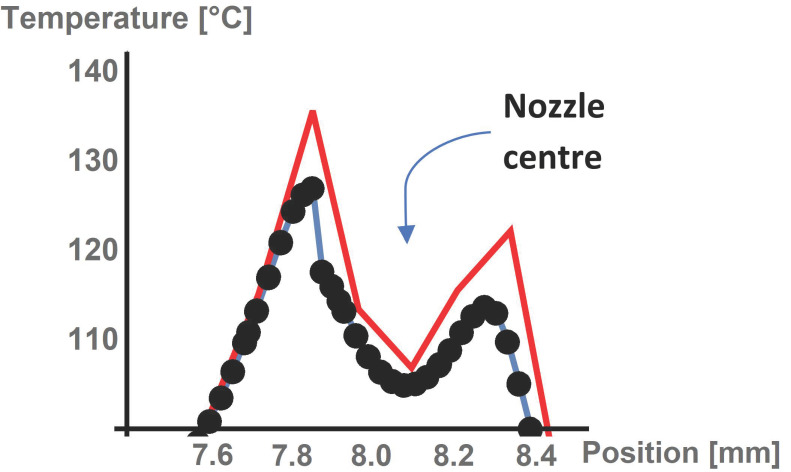
Mesh convergence evaluation under the nozzle. Black—extremely fine Mesh; blue—fine mesh; red—coarse mesh.

**Figure 4 polymers-13-04213-f004:**
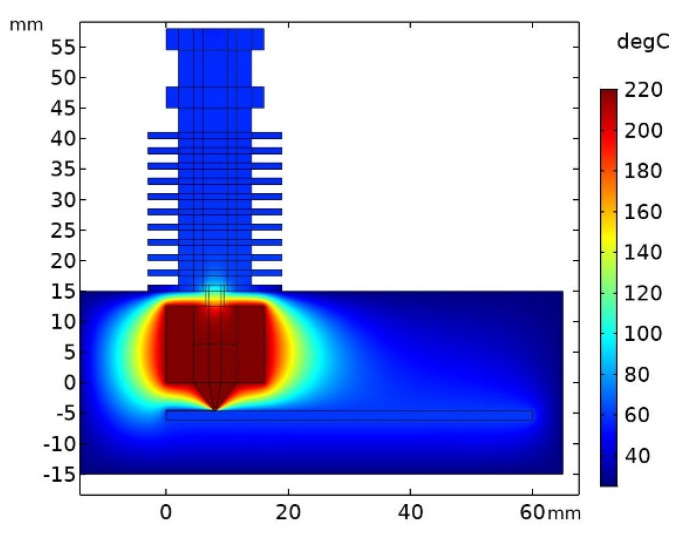
Static model output—temperature distribution in the examined area.

**Figure 5 polymers-13-04213-f005:**
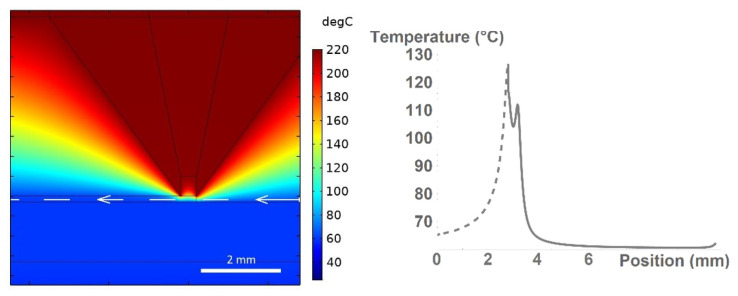
Detail of the printer nozzle/bed interface (left). Temperature distribution along the white line, where the white arrow denotes the direction of the *x*-axis (right). Printhead moves in the right direction.

**Figure 6 polymers-13-04213-f006:**
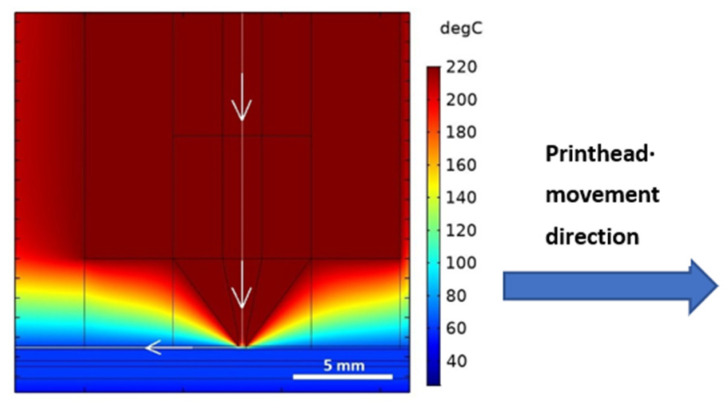
Temperature distribution at movement speed of 5 mm/s and at the time *t* = 5 s. Temperature distribution along the white line, where the white arrow denotes the direction of the *x*-axis is shown in Figure 7.

**Figure 7 polymers-13-04213-f007:**
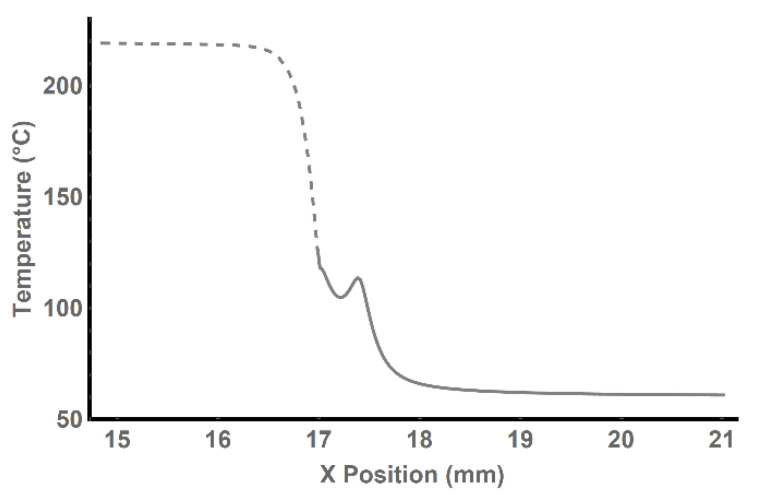
Temperature distribution in the material during printing (nozzle outlet is located on position 17.3 on the *x*-axis).

**Figure 8 polymers-13-04213-f008:**
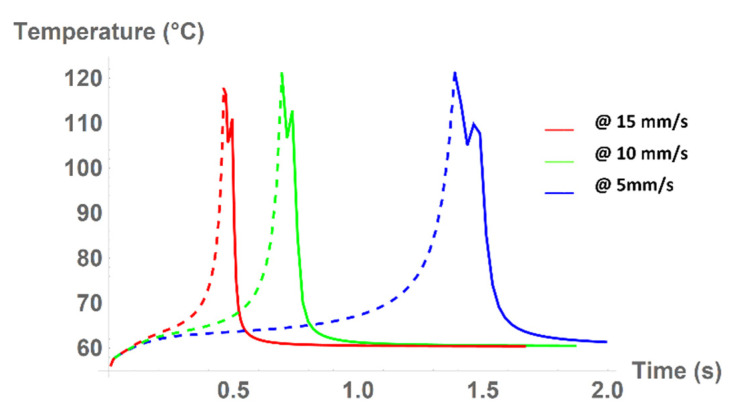
Temperature of the fixed-point during printing.

**Table 1 polymers-13-04213-t001:** Boundary conditions of the static model.

Edge Denotation	Boundary Condition
Problem boundary	25 °C
Inner part of the nozzle	220 °C
External part of the extruder	220 °C
Printer bed	60 °C
Outer part of the extruder (heat sink)	Power dissipated into surroundings (Σ40 W)

**Table 2 polymers-13-04213-t002:** Material parameters of the static model.

Model Area	Material	Color	Thermal Conductivity W·m^−1^·K^−1^	Specific Heat Capacity J·kg^−1^·K^−1^
Printer bed	Steel	Purple	76	440
Printer nozzle	Brass	Yellow	120	400
Heat sink body	Aluminum	Darker turquoise	238	900
Extruder body				
Printing material	PLA	Dark green	0.13	2100
Surroundings	Air	Grey	0.02	1.01
Teflon tube	Teflon	Lighter turquoise	0.24	1050

**Table 3 polymers-13-04213-t003:** Parameters for mesh quality check and computational time.

Name	Max. Triangle Element Size (mm)	Min. Triangle Element Size (mm)	Comp. Time (s)	Number of Mesh Elements	Number of Degrees of Freedom
Ex. Fine	0.05	0.025	288	101,740	214,341
Fine	0.1	0.05	246	73,386	157,449
Coarse	1	0.5	31	5536	21,297

**Table 4 polymers-13-04213-t004:** Parameters of time-dependent solver.

Speed (mm/s)	Stop Time (s)	Time Step (s)	Num. of Values (Time)
5	3	0.025	120
10	2.5	0.020	120
15	1.6	0.008	200

**Table 5 polymers-13-04213-t005:** Comparison of cooling processes of the material for various movement speeds of the printer head.

Movement Speed	Maximum “Second Tooth” Temperature (°C)	Minimum Temperature (°C)	Calculated Slope (°C/s)
5 mm/s	109.7	60.5	−201.7
10 mm/s	112.7	60.5	−404.1
15 mm/s	111.1	60.5	−620.2

## Data Availability

The data presented in this study are available on request from the corresponding author.

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
