# Peer review of "Mathematical Modelling of Temperature Distribution in Selected Parts of FFF Printer during 3D Printing Process"

_polymers, 2021, doi:10.3390/polym13234213_

Round 1

Reviewer 1 Report

The paper is round but needs some slight improvements in the presentation.

line 50: 'process are'

line 119: give the full name for BDF

ad Fig. 2 and related text: explain in which direction the movement is going and how it is realised with this mesh in simulation

ad Fig. 3: how are the positions on the x-axis are related to the model/mesh?

line 143: 'coarse'

line 147: '0.05 mm'

Table 3: give better representation of the titles in the table, especially 'Num.' and 'Name'. What does 'Name' mean here?

line 192: 'deposited layer'

ad Fig. 6: how are the positions on the x-axis are related to the model/mesh?

ad Fig. 7: what is the direction of movement?

ad Fig. 8: how are the positions on the x-axis are related to the model/mesh?

ad Fig. 9: ad a legend for the printing speeds

ad Table 5: add a discussion of Table 5, especially the slopes in the text

Author Response

Dear reviewer, thank you for your suggestions and valuable feedback. It helped us to reveal the ambiguities that were present in the original manuscript and thus the article has gained a significantly better level and at the same time is more illustrative for readers. The changes in manuscript are highlighted in yellow colour.

Reviewer 2 Report

The authors developed an model to analyse the temperature of component of the equipment during the fused filament fabrication process. There are several issues in the manuscript that should be addressed.

1. Suggest the authors to use ISO/ASTM terminology to describe the process. For example, FFF should be an material extrusion process.

  • Vassilolakas et al. (2021), Fabrication of parts with heterogeneous structure using material extrusion additive manufacturing, Virtual and Physical Prototyping 16(3), 267-290
  • Yap et al. (2020), A review of 3D printing processes and materials for soft robotics, Rapid Prototyping Journal 26(8), 1345-1361

2. There has been numerous work done on thermal simulation of material extrusion AM process, and yet, the authors have not provide a review on existing work and what new contributions arise from this work.

  • Moretti et al. (2021), In-process simulation of the extrusion to support optimisation and real-time monitoring in fused filament fabrication, Additive Manufacturing 38, 101817
  • Luis et al. (2020), 3D Direct Printing of Silicone Meniscus Implant Using a Novel Heat-Cured Extrusion-Based Printer, Polymers 12(5), 1031

3. For the parameters used in the simulation, for example the speed of the print head, are they corresponding to the values of actual process?

4. How would the results obtained from this model benefit the fabrication process? Is there adjustment needed for the nozzle temperature etc? Any validation done with actual prints?

Author Response

Dear reviewer, thank you for your suggestions and valuable feedback. We improved the original manuscript and thus the article has gained a significantly better level and at the same time is more illustrative for readers. The required changes in manuscript are highlighted in green colour.

Reviewer 3 Report

This paper modeled and discussed the temperature distribution of the 3D printing system including the nozzle, heat sink, PLA filament and the printing bed with the help of FEM analysis. Both static and dynamic scenarios are discussed and the temperature evolution of the filament before/after the extruding from the nozzle was discussed in detail. The classical heat conduction equation was used to describe the whole system, which should be reasonable to solve static temperature distribution. My mainly interest is how the authors build the dynamic model.  Unfortunately, the authors didn’t put many words on how their dynamic model was built and solved in detail, and what I can read from this paper is that they are complicated. For modeling time dependent dynamic model, the solving process could be very tricky since it coupled with structure/mesh reconstruction. During the real 3D printing process, the micro polymer extruder drives the PLA filament and squeeze it through the hot nozzle, the filament feeding rate must match the movement speed of the nozzle in order to produce a consistent flow rate of the melted polymer. This feeding movement creates a mass transfer process that coupled to the existing heat conduction solver, and it will definitely change the temperature distribution inside the nozzle. The boundary condition choice in the model described in this paper also need to be carefully tuned to be able to capture the mass movement between the filament and the inner wall of the nozzle. I hope the detailed boundary should be clearly labeled in figure 1 instead of a rough description in table 1. A clearly labeled boundary condition will be helpful for people who want to reproduce your calculating results.

Another issue is modeling the correct morphology of the extruded molten PLA. Inside the nozzle, there is a huge internal pressure built inside the material during the extruding and melting process, when the molten PLA is squeeze out of the nozzle, it is usually shown as a droplet under the surface tension and the release of the internal pressure will expand the volume of the squeezed molten PLA and form a expended globule in front of the nozzle. The tip of the nozzle will be emerged inside this molten globule and shave the front globule into a flat surface. To capture this dynamic process, 2D FEM calculation may not accurate enough and it could potentially lead to an inaccurate description on the temperature distribution such as the secondary peak mentioned in the paper. Instead of looking at a complete 3D printing system built with an inaccurate model, I would prefer seeing the author could address one region with the detailed and accurate model, which would be much helpful for people in 3D printing industry.

I hope the authors could address those details mentioned above and make some major changes before sending revised manuscript to the journal office. Or try some other journals instead.

Author Response

Dear reviewer, thank you for your suggestions and valuable feedback. We are aware of some drawbacks our simulation has. We believe that the article is now significantly improved, not just in terms of higher clarity for the readers. Boundary condition for the models are in manuscript highlighted in turquoise colour, other changes are highlighted in yellow or green.

Round 2

Reviewer 2 Report

Suggest the authors to conduct experiments to validate the simulations and to discuss how it affects the 3D printing process.